# Social capital and cognitive decline: Does sleep duration mediate the association?

**Liqun Wang[1], Jiangping Li[1], Zhizhong Wang[2]\*, Yong Du[3,4], Ting Sun[3], Li Na[3], Yang Niu[5,6]\***

**1** Department of Epidemiology and Statistics, School of Public Health and Management at Ningxia Medical University, Yinchuan, China, **2** Department of Epidemiology and Statistics at School of Public Health of Guangdong Medical University, Dongguan, China, **3** Surgical Laboratory of General Hospital, Ningxia Medical University, Yinchuan, China, **4** School of Clinical Medicine at Ningxia Medical University, Yinchuan, China, **5** Key Laboratory of the Ningxia Ethnomedicine Modernization, Ministry of Education, Ningxia Medical University, Yinchuan, China, **6** School of Traditional Chinese Medicine, Ningxia Medical University, Yinchuan, China

\* wzhzh_lion@126.com (ZW); niuyang0227@163.com (YN)

## Abstract

### Background

Studies have found that social capital (SC) is associated with the risk of cognitive decline; however, the mechanism explaining how SC leads to cognitive decline is unclear. The current study examines the mediation effect of sleep duration on the relationship between SC and cognitive decline in Chinese older adults.

### Methods

A cross-sectional study of 955 community-dwelling aged 60 or over was conducted. The mini-mental state examination (MMSE), self-report sleep duration questionnaire, and social capital scales were administered during the face-to-face survey. The Bootstrap methods PROCESS program is employed to test the mediation model.

### Results

After controlling for covariates, both social cohesion and social interaction were positively correlated with the MMSE score (p<0.001), and social cohesion was negatively correlated with sleep duration (p = 0.009); On the contrary, sleep duration was negatively correlated with MMSE score (p<0.001). Linear regression analysis showed social cohesion was positively associated with the MMSE score (β = 0.16, p = 0.005), while sleep duration was associated with an increased risk of cognitive decline (β = -0.72, p<0.001). Sleep duration has mediated the relationship between social cohesion and cognitive decline (explaining 21.7% of the total variance).

### Conclusions

Social capital negatively associated with the risk of cognitive decline in this Chinese population, and sleep duration may partly explain this relationship. It may be a suggestive clue to

**Data Availability Statement:** All relevant data are within the paper and its Supporting Information files.

**Funding:** The study was supported by the Research and Development Plan of the 13th five-year plan of Ningxia autonomous region (the major S&T projects.) (grant number 2016BZ02) and the National Natural Science Foundation of China (grant number 81860599).

**Competing interests:** The authors have declared that no competing interests exist.

identify those at a higher risk of progressing to cognitive impairment. Further prospective study in need to confirm this finding due to the cross-sectional design.

## Introduction

Expanded longevity is one of the most remarkable success stories in human history, and this also directs the population aging. The proportion of people aged 60 years and older is expected to rise to 22% of the total population in the coming decades, which is from 800 million to 2 billion [1]. One of the consequences of rapid population aging is the increased prevalence of aging-related diseases, of which dementia and mild cognitive impairment were the commonly prevalent neuropsychiatric disorders in older adults [2].

The determinants of cognitive decline include biological (Apolipoprotein E, protein tau, β-amyloid) and environmental factors (education, lifestyle, diet, medical service, social capital, etc.) [3–5]. Studies have also reported that sleep duration is an independent neurobehavioral predictor of cognitive disorders [6]. A meta-analysis suggested that longer sleep duration was associated with higher risks of mild cognitive impairment [7]. Another study revealed a U-shaped association that showed either shorter or longer sleep duration was associated with a higher risk of cognitive decline [8].

Social capital (SC) is a characteristic of social life, including interpersonal trust, norms of reciprocity, mutual aid, and social involvement (like socializing with friends, relatives, colleagues, or neighbors) [9], which has linked with several beneficial health outcomes [10, 11]. As a social determinant of health, SC may play an essential role in protecting individuals from cognitive decline. At least one study has found that SC accrued in early and midlife may reduce the detrimental influences of psychological stress on cognitive functioning in later life [12]. Specifically, possessing a rich SC (supportive personal network with numerous types of relationships, e.g., neighbor and friends) has been associated with better cognitive function among the old adults [13–17]. One possible explanation is individuals with rich SC lead to less life stress and more leisure time [12]. Also, one study found that poor SC might increase the risk of cognitive decline among elderly residents in Wuhan, China [18].

Besides, SC has been associated with sleep duration [19, 20]. A notable exception is one study that revealed an inverted U-shaped association between SC and sleep duration [19]. Takahashi et al. [21] reported that Japanese workers who had higher neighborhood or workplace social capital had a better quantity of sleep (not too short and too long).

There still unclear how SC leads to better cognitive function in older individuals. A previous study reported that shorter sleep duration mediated the association between homocysteine and cognitive decline [22]. They argued that short sleep duration might cause an increased homocysteine level, then strengthened Aβ accumulation, which is a critical pathological process of cognitive decline. The present study sought to examine the mediating effect of sleep duration on the relationship between SC and cognitive decline in old Chinese adults. We hypothesized that rich SC is associated with better cognitive function and that this association would be mediated at least in part by appropriate sleep duration.

## Methods and materials

### Study sample

Data were abstracted from a cross-sectional survey conducted from April 2017 to July 2017 at Ningxia province, China. The detailed sampling process can be found elsewhere [23]. Here, in

summary, the participants were selected using a multi-stage sampling method: firstly, four counties were selected from a total of 22 counties in the province using a stratified sampling design according to the proportion of the minority population and the economic status. Secondly, twenty rural communities and twenty urban communities were selected among the four selected counties (with a total of 166 urban communities and 628 rural communities) using random sampling methods. Thirdly, 115 households were selected in each target communities using a systematic sampling method. Finally, one eligible family member from each household was determined to attend the survey using the Kish table. There were 615 households not responded after three times attempt to contact, and 3,985 eligible participants were finally selected. Of them, 1159 participants were aged 60 and over, and 104 participants were excluded due to the cognitive function test missing. Of them, 1159 participants were aged 60 and over, and 104 participants were excluded due to the cognitive function test's missing value. Ultimately, 955 participants were included in this study (**Fig 1**). The inclusion criteria are a) living at the present address for at least six months and b) aged 60 years or older. The exclusion criteria were the following: a) unconsciousness caused by any forms; b) the acute phase of a cerebrovascular accident; c) a severe illness (e.g., stroke, cerebral infarction or myocardial infarction) that prevents communication; d) any obvious cognitive disabilities or deafness, aphasia or other language barriers; e) people reported with depression (by asking "do you ever be told have depression by your doctors") and f) with sleep disorders and taking hypnotics or psychotropic medications, as well as some particular occupation need to going to bed late.

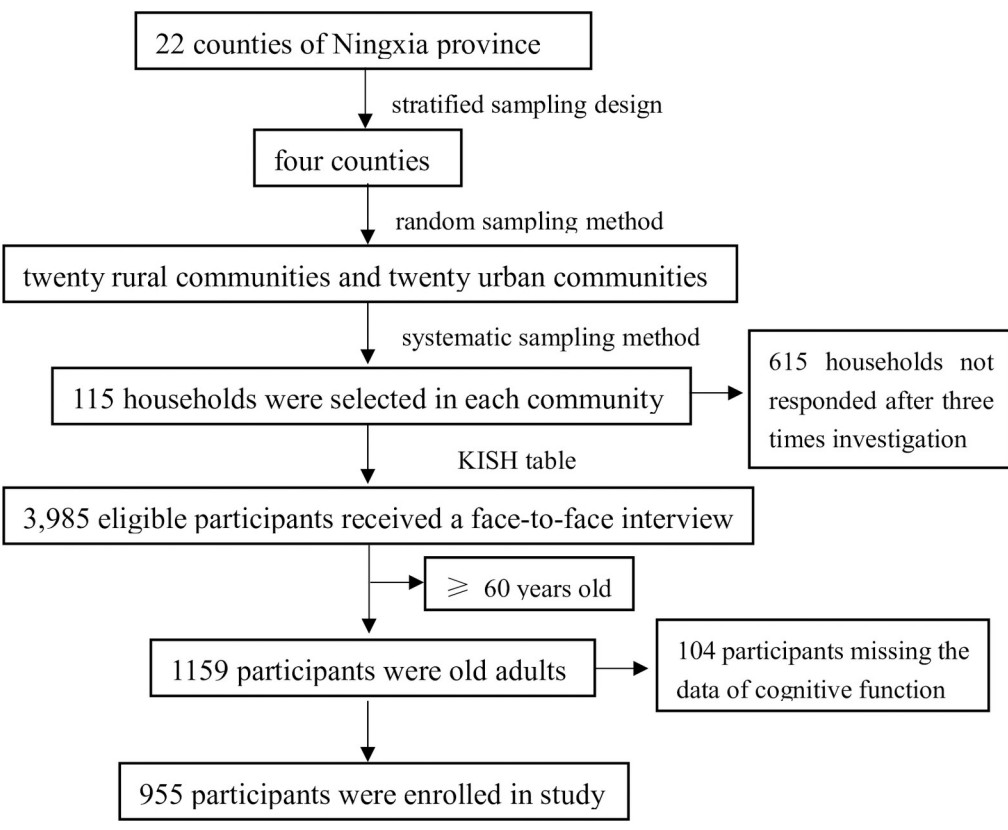

**Fig 1. Participant screen process.**

The Institutional Review Board of the General Hospital of Ningxia Medical University (approval number 2017–200) approved this study. All the participants provided a written consent form at the beginning of the survey.

## The field process

The trained medical students served as investigators. With local community leaders' cooperation, the investigators visited the participants' houses to guide them to finish the survey, then described our study and questionnaire. Under the participants' agreement, our investigators read the questions one by one to them and then recorded their answers, and the survey lasted approximately 45 minutes. As for respondents with low mini-mental state examination (MMSE) scores, other information is provided by family members. The finished questionnaire was double-checked immediately by a separate supervisor in the field.

## Measurement

**Cognitive function.** The Chinese version of the MMSE scale was employed to assess the cognitive function, which has high sensitivity (90.8%) and specificity (93%) for screening cognitive disorders [24]. The MMSE consists of 19 questions to measure the five different domains of cognitive function (1) orientation, (2) memory, (3) attention and calculation, (4) language, and (5) constructional praxis (coping task, e.g. copying intersecting pentagons). And has a total score ranging from 0 to 30; a higher score reflects better cognitive function. We categorified cognitive performance into two categories (cognitive decline vs. normal according to Cui et al. suggested criteria [25]: MMSE ≤17 for those with no formal education; MMSE ≤ 20 for those with primary school education (≥6 years); and MMSE≤24 for those with junior high school education or above (≥ 9 years).

**Social capital.** SC was evaluated using the social capital scale to cover the two dimensions of social capital (social cohesion and social interaction) developed by Mujahid [26]. The social cohesion subscale consists of 4 items: 1) People around here are willing to help their neighbors, 2) People in my neighborhood generally get along with each other, 3) People in my neighborhood can be trusted, 4) People in my neighborhood share the same values. Each item ranged from 1 to 5 (1 = strongly disagree, 2 = disagree, 3 = neutral, 4 = agree, and 5 = strongly agree). Cronbach's alpha was 0.88 among the Chinese sample [27], and among this sample was 0.81. The social interaction scale consists of five items: 1) You and other people in the community (village) help each other (e.g., look after children, help buy something and borrow tools), 2) When a neighbor is not at home or going out, you can help him look after the house or property, 3) people in the community (village) talk about each other's personal matters (children care, exercise, etc.), 4) participate in group activities together with people in the community (village), and 5) communicate with each other on the street. Each item scored from 1 to 4 in response to a 4-point Likert scale (from never to often). The previous study has shown the Chinese version of the social interaction scale has good reliability and validity [27]. The Cronbach's alpha in this sample was 0.76.

**Sleep duration.** Three questions "What time do you usually go to sleep at night?", "What time do you usually rise in the morning?" and "In general, do you take afternoon nap often?". The time between bedtime and rise up as the crude sleep duration, then it adjusted depends on the response to the question "In general, do you take afternoon nap often?" if the answer is yes, then the modified sleep duration is crude sleep duration plus one hour.

Socio-demographic information includes age, gender (male vs. female), ethnicity (Han vs. minority), residence (rural vs. urban), educational attainment (continuous data), marital status (married vs. unmarried/widowed/divorced), family income (as measured by the self-reported

family average individual income per month and the answer included five groups: <1,000 RMB, 1,000–1,999 RMB, 2,000–2,999 RMB, 3,000–4,999 RMB, and 5,000 RMB or more) were collected using a standard form.

The data about body mass index (BMI = weight (kg)/height (m)$^2$), fasting blood glucose (mmol/L), smoking (defined as at least one cigarette per day and last for six months or more), alcohol use (defined as a drink of at least one glass of alcohol, that equals 1/2 bottle of beer or 125-milliliter grape wine or fruit wine or 40-milliliter white wine, in a day for the past 12 months), hypertension (yes vs. no), dyslipidemia (yes vs. no) were abstracted from the medical record.

## Statistical analyses

Analyses were performed using the Statistical Package for the Social Sciences (SPSS) version 24.0 (IBM Inc., Chicago, Illinois, USA). We performed a logarithmic transformation of the MMSE score to fit the normal distribution. Means and standard deviations (SD) or were used to describe continuous variables; counts and proportions were used to describe categorical variables. The bivariate test using Student's t-test or Chi-square test. Partial correlations were employed to create the correlation matrix under controlling the socio-demographic covariates. Three separate linear regression models were performed to examine the association among SC, sleep duration, and MMSE. In summary, model l include the SC and sleep duration; model 2 adjusted with covariate variables (age, gender, ethnicity, residence, educational attainment, marital status, family income, body mass index, smoking, alcohol use, fasting blood glucose, hyperlipidemia, hypertension); and the interaction between SC and sleep duration were added in model 3. And collinearity detected where the tolerance >0.1 and Variance Inflation Factor (VIF) <5 indicate multicollinearity can be ignored [28]. Bootstrap methods of PRO-CESS developed by *Hayes* was employed to test the mediation effect of sleep duration on the relationship between SC and MMSE score [29]. The bias-corrected percentile bootstrap confidence interval does not contain 0 indicate that the mediation effect statistically significant [30]. Sensitivity analyses were performed using Structural Equation Modelling (SEM) approach.

## Results

### Demographic characteristics of participants

The demographic characteristics of the participants were shown in **Table 1**. The average age was 66.9 (SD = 5.3) years, with a range of 60 to 80 years. Slightly about half (46.7%) were male, the mean educational attainment was 3.7 (SD = 4.2) years, and about 81.9% were married. The mean sleep duration was 8.1 (SD = 1.5) hours, the mean score of social cohesion was 15.7 (SD = 2.5), and for the social interaction was 13.2 (SD = 3.7). The mean score of MMSE was 23.6 (SD = 5.4). And the prevalence of cognitive decline (CD) was 15.8%. Participants with CD were older, more likely to have longer sleep duration, and lower SC scores than those with normal cognitive performance.

### The binary correlation matrix

The partial correlation matrix showed in **Table 2**. After controlling for socio-demographic variables (age, gender, ethnicity, residence, educational attainment, marital status, family income) and health variables (BMI, smoking, alcohol use, FBG, hyperlipidemia, hypertension), both the social cohesion (r = 0.12, P<0.001) and social interaction (r = 0.09, P<0.05) were positively correlated with MMSE score; the sleep duration was negatively correlated with MMSE score (r = -0.24, P<0.001); and the social cohesion was negatively correlated with sleep duration (r =

**Table 1. Demographic characteristics of participants (n = 955).**

| Variables | Total (N = 955) | cognitive decline (N = 151) | normal (N = 804) | $\chi^2$/t | p |
|---|---|---|---|---|---|
| Age, mean (SD), years | 66.9(5.3) | 68.7(5.8) | 66.6(5.1) | 4.08[a] | <0.001 |
| Gender, male, n (%) | 446(46.7) | 36(23.8) | 410(51.0) | 37.66[b] | <0.001 |
| Ethnicity, han, n (%) | 646(67.6) | 109(72.2) | 537(66.8) | 2.83[b] | 0.243 |
| Residence, rural, n (%) | 479(50.2) | 88(58.3) | 391(48.6) | 4.73[b] | 0.030 |
| Marital status, n (%) | | | | 3.10[b] | 0.078 |
| unmarried/widowed/divorced | 173(18.1) | 35(23.2) | 138(17.2) | | |
| Married | 782(81.9) | 116(76.8) | 666(82.8) | | |
| Educational attainment, mean (SD), years | 3.7(4.2) | 1.6(3.7) | 4.0(4.1) | 7.27[a] | <0.001 |
| Family income, n (%) | | | | 37.95[b] | <0.001 |
| <1000 RMB | 505(52.9) | 111(73.5) | 394(49.0) | | |
| 1000~1999 RMB | 192(20.1) | 17(11.2) | 175(21.7) | | |
| 2000~2999 RMB | 148(15.5) | 14(9.3) | 134(16.7) | | |
| 3000~4999 RMB | 87(9.1) | 6(4.0) | 81(10.1) | | |
| ≥5000 RMB | 23(2.4) | 3(2.0) | 20(2.5) | | |
| Smoking, n (%) | 181(19.0) | 18(11.9) | 163(20.3) | 5.77[b] | 0.016 |
| Alcohol use, n (%) | 126(13.2) | 5(3.3) | 121(15.0) | 15.29[b] | <0.001 |
| BMI, mean (SD) | 25.8(7.9) | 25.5(4.0) | 25.8(8.3) | 0.44[a] | 0.662 |
| FBG, mean (SD) | 5.1(1.6) | 5.0(1.4) | 5.1(1.6) | 0.53[a] | 0.599 |
| Hyperlipidemia, n (%) | 38(4.0) | 3(2.0) | 35(4.4) | 1.86[b] | 0.172 |
| Hypertension, n (%) | 385(40.3) | 60(39.7) | 325(40.4) | 0.02[b] | 0.874 |
| Social cohension, mean (SD) | 15.7(2.5) | 15.3(2.8) | 15.7(2.5) | 1.89[a] | 0.059 |
| Social interaction, mean (SD) | 13.2(3.7) | 12.5(4.3) | 13.3(3.6) | 2.57[a] | 0.010 |
| Sleep duration, mean (SD), hours | 8.1(1.5) | 9.0(1.6) | 7.9(1.4) | 7.58[a] | <0.001 |

SD: standard deviation; BMI: body mass index; FBG: fasting blood glucose

a: t-test used

b: chi-square test used.

-0.11, P = 0.003). As shown in **Fig 2**, a U-shaped association between sleep duration and MMSE score was found.

## Linear regression analysis

As **Table 3** showed, social cohesion was positively associated with the MMSE score; namely, those with rich social cohesion may predict better cognitive performance. In contrast, sleep

**Table 2. Correlation matrix between SC, sleep duration, and MMSE score (n = 955)[a].**

| | Mean | SD | 1 | 2 | 3 | 4 |
|---|---|---|---|---|---|---|
| 1.MMSE | 23.6 | 5.4 | 1 | | | |
| 2.Social cohension | 15.7 | 2.5 | 0.12** | 1 | | |
| 3.Social interaction | 13.2 | 3.7 | 0.09* | 0.46** | 1 | |
| 4.Sleep duration | 8.1 | 1.5 | -0.24** | -0.11* | -0.05 | 1 |

**p<0.001

*p<0.05, SD = standard deviation

a: The covariates include age, gender, ethnicity, residence, educational attainment, marital status, family income, BMI, smoking, alcohol use, FBG, hyperlipidemia, hypertension were adjusted using Partial correlation method.

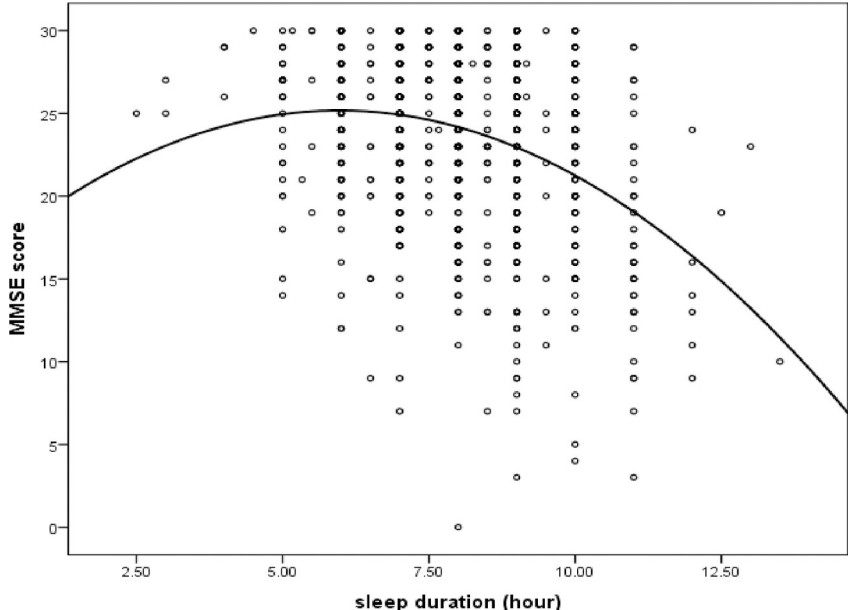

**Fig 2. A U-shaped association between sleep duration and MMSE score.**

duration was inversely associated with the MMSE score (**model 2**). The same was true for social interaction. The association between SC and MMSE score disappeared when adding the interaction between SC and sleep duration (model 3), indicating that sleep duration is a possible mediator.

## Mediation effect of sleep duration on the relationship of SC and cognitive decline

As shown in **Table 4**, after controlling for covariates, there is a significant mediation effect of sleep duration on the relationship between social cohesion and cognitive decline. The results showed that both the direct effect (p = 0.004) and the indirect effect (p = 0.002) were significant. The mediation effect explained 21.7% (0.045/0.207) of the total variance. No mediation effect of sleep duration in the relationship between social interaction and cognitive decline was found.

**Table 3. Linear regression model for interaction between SC and sleep duration on cognitive decline (n = 955).**

| Variables | Model 1 | | Model 2 | | Model 3 | |
|---|---|---|---|---|---|---|
| | P value | β (95%CI) | P value | β (95%CI) | P value | β (95%CI) |
| Social cohesion | 0.192 | 0.09(-0.04,0.22) | 0.005 | 0.16(0.05,0.27) | 0.522 | 0.23(-0.39,0.84) |
| Sleep duration | <0.001 | -0.99(-1.21,-0.77) | <0.001 | -0.72(-0.91,-0.52) | 0.311 | -0.60(-1.75,0.55) |
| Social cohesion×sleep duration | NA | NA | NA | NA | 0.900 | -0.01(-0.08,0.07) |
| Social interaction | 0.001 | 0.14(0.06, 0.23) | 0.020 | 0.09(0.01,0.16) | 0.607 | 0.11(-0.31,0.52) |
| Sleep duration | <0.001 | -0.98(-1.20,-0.76) | <0.001 | -0.74(-0.93,-0.54) | 0.037 | -0.71(-1.37,-0.05) |
| Social interaction×sleep duration | NA | NA | NA | NA | 0.950 | -0.01(-0.05,0.05) |

Model l = SC+ sleep duration; Model 2 = Model l + covariate variables (age, gender, ethnicity, residence, educational attainment, marital status, family income, body mass index, smoking, alcohol use, fasting blood glucose, hyperlipidemia, hypertension); Model 3 = Model 2 + interaction between SC and sleep duration; Social interaction and social cohesion were tested separately in all the models. β: beta; 95%CI: 95% confidence interval; NA: not applicable.

$R^2$ for social cohesion in model 1–3 are 0.081, 0.389, 0.389 respectively; $R^2$ for social interaction in model 1–3 are 0.089, 0.388, 0.389 respectively.

**Table 4.  The mediating effect of sleep duration on the relationship between SC and cognitive decline [*].**

| Effect | | | | Bias-Corrected 95%CI | |
|---|---|---|---|---|---|
| | *β* | *SE* | *P*-value | Lower | Upper |
| Social cohesion | | | | | |
| Total effect | 0.207 | 0.058 | <0.001 | 0.093 | 0.332 |
| Indirect Effects | 0.045 | 0.016 | 0.002 | 0.018 | 0.082 |
| Direct Effects | 0.162 | 0.057 | 0.004 | 0.050 | 0.275 |
| Social interaction | | | | | |
| Total effect | 0.104 | 0.039 | 0.007 | 0.026 | 0.181 |
| Indirect Effects | 0.004 | 0.002 | 0.133 | -0.001 | 0.009 |
| Direct Effects | 0.090 | 0.038 | 0.019 | 0.015 | 0.165 |

[*] After controlling for age, gender, ethnicity, residence, educational attainment, marital status, family income, BMI, smoking, alcohol use, FBG, hyperlipidemia, hypertension.

Considering the possible U-shape relationship between sleep duration and cognitive decline, the exploring analysis conducted stratified (use a cutoff point of mean sleep duration of the total sample) by sleep duration showed in **Table 5**. The mediation effect of sleep duration on the relationship between social cohesion and cognitive decline persists in those who had eight hours and longer sleep duration. However, the mediation effect disappears in those who had less than eight hours of sleep duration per day.

## Sensitivity analysis

As showed in **Fig 3**, the SEM approach was peformed to sensitivity analysis. The results showed social cohension was positively associated with cognitive function and inversely related to sleep duration, and sleep duration negatively associated with cognitive function. These results was consistent with the binary correlation results even though the pathway coefficient was not the same due to the different methods. This validated the methodological robustness of findings.

## Discussion

The current study examined the association between SC and cognitive decline. And provides the primary evidence for the relationship between SC and cognitive decline among older

**Table 5.  The mediation model of social cohesion stratified by sleep duration[a].**

| Effect | | | | Bias-Corrected 95%CI | |
|---|---|---|---|---|---|
| | *β* | *SE* | *P*-value | Lower | Upper |
| sleep duration <8 h | | | | | |
| Total effect | 0.196 | 0.093 | 0.038 | 0.011 | 0.380 |
| Indirect Effects | -0.001 | 0.008 | 0.954 | -0.012 | 0.022 |
| Direct Effects | 0.197 | 0.093 | 0.038 | 0.011 | 0.380 |
| sleep duration ≥8 h | | | | | |
| Total effect | 0.164 | 0.074 | 0.026 | 0.019 | 0.309 |
| Indirect Effects | 0.041 | 0.021 | 0.025 | 0.008 | 0.091 |
| Direct Effects | 0.123 | 0.072 | 0.089 | -0.019 | 0.265 |

a: The effect adjusted the covariates include age, gender, ethnicity, residence, educational attainment, marital status, family income, BMI, smoking, alcohol use, FBG, hyperlipidemia, hypertension.

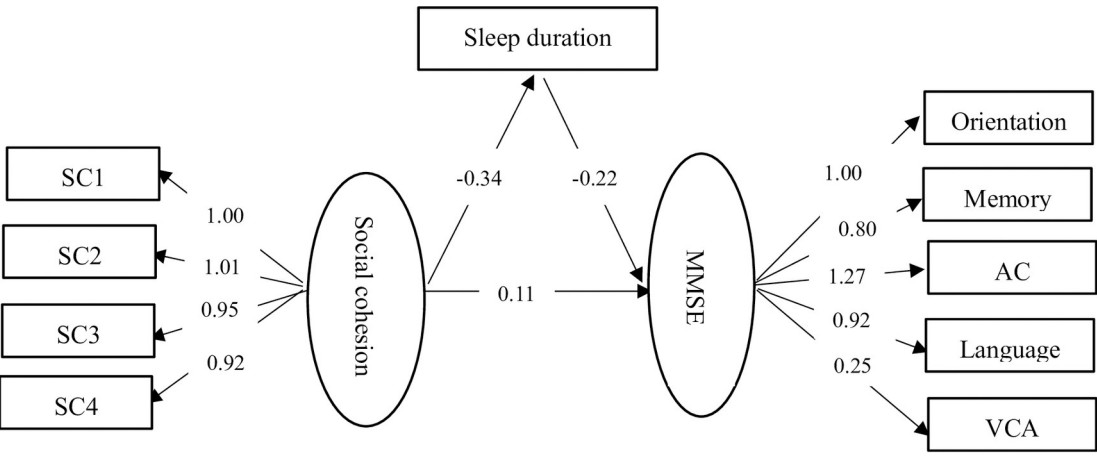

**Fig 3.**

adults, suggesting a possible mechanism to explain how SC is related to reduce cognitive impairment in studies examining that association. As hypothesized, the results revealed a positive association of SC with cognitive performance in older Chinese adults, and this relationship was partially mediated by sleep duration. The mediation effect is accounting for 21.7% of the total effect in the total sample, while no mediation effect was found among those who had less than eight hours of sleep duration.

Sleep duration was negatively associated with MMSE. Long sleep duration might be a risk factor for cognitive decline, consistent with prior research that found long sleep duration was associated with cognitive impairment [31–33]. A cohort study found long sleep duration was associated with poorer cognitive performance among adults aged 41–75 years [34]. Previous studies also have reported that long sleep duration is associated with low MMSE scores among old adults [35–39]. One possible explanation is increased sleep fragmentation associated with decreased cognitive performance, and so it may be that longer sleep duration may emerge more frequent nighttime wakes or sleep in bed much more time [40]. Additionally, obstructive sleep apnea syndrome (OSAS) can cause deterioration in cognitive functions, and the study reported that OSAS was more prevalent in extended sleep duration groups [41, 42]. Besides, long sleep duration has been associated with an increased level of inflammatory factors [43, 44], and elevated inflammatory cytokine levels increase the risk of cognitive decline [45]. Furthermore, reports revealed a U-shaped association with a higher risk of cognitive decline in older adults with either short or long sleep duration [7, 46, 47].

Compared with lower SC, the average sleep duration was shorter in those who have higher SC. This finding was consistent with the previous study that found lower neighborhood SC was negatively associated with short sleep duration among Japanese male adults [17]. China is a typical *Guanxi*-based society, and the previous study has reported that *Guanxi* (traditional Chinese social interaction) has almost the same connotation as social capital [48]. Additionally, rural China's culture values trust, mutual assistance, and reciprocal exchange, which provide cultural soil for cultivating social capital [49].

The current study also found SC was positively correlated with the MMSE score and might be a protective factor of cognitive function. One study conducted in Taiwan revealed that increased social support is associated with better cognitive function in older adults [50]. Also, Holtzman and his colleagues reported a positive association between social support and

cognitive function among older adults [51]. Moreover, a three-year cohort study found that social networks reduced the incidence of dementia in older adults [52]. On the one hand, the possible mechanisms were social activities provide the challenge of effective communication and participation in complex interpersonal exchanges [53]. On the other hand, emotional support might buffer against physiological stress and benefit cognitive function [54]. Furthermore, a recent study manifested that social capital could help older adults continue to independently live in local communities and handle life stressors efficiently, even when they encounter declines in their physical and cognitive health. Social capital can provide older residents with a sense of security and belonging and is an important reserve domain in old age [55].

### Strength and limitations

Given the increasing rate of aging and the incidence of dementia in the elderly Chinese population, the present findings have relevance for understanding the mechanisms of how SC is linked with cognitive function. And provide primary evidence for developing interventional program for cognitive decline in minority areas. Several limitations were identified; first, the cross-sectional design prevents making causal inferences from the relationships between SC and cognitive decline reported here. Hence, further longitudinal design would be necessary to determine causal relationships in the future. Second, potential confounding variables like depression that were not included in the analysis may lead to overestimation of the association between cognitive function and sleep duration. Third, due to the feasibility consideration, bedtime and sleep duration were collected via a self-reported survey question; it may involve information bias even though it has been found to have a reasonable correlation with actigraphic measurement [56].

### Conclusions

Social cohesion, one dimension of the social capital, positively associated with cognitive function, and sleep duration partly mediated this relationship. The findings provides the primary evidence for better understanding how SC is related to reduce risk of cognitive decline in studies examining that association. Hence, clinicians can suggest patients communicate more with others (chat, play chess or exercise together, etc.) to improve the social capital, and in turn maintain their cognitive function.

### Supporting information

**S1 Data.**
(SAV)

### Author Contributions

**Conceptualization:** Liqun Wang, Zhizhong Wang, Li Na, Yang Niu.

**Data curation:** Liqun Wang, Jiangping Li, Zhizhong Wang, Yong Du, Ting Sun.

**Formal analysis:** Liqun Wang, Jiangping Li, Zhizhong Wang.

**Supervision:** Zhizhong Wang, Yang Niu.

**Writing – original draft:** Liqun Wang, Zhizhong Wang.

**Writing – review & editing:** Jiangping Li, Zhizhong Wang.

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
