## [Decision Letter · Decision Letter 0]

11 Dec 2020

PONE-D-20-28198

Social capital and cognitive decline: does sleep duration mediate the association?

PLOS ONE

Dear Dr. Wang,

Thank you for submitting your manuscript to PLOS ONE. After careful consideration, we feel that it has merit but does not fully meet PLOS ONE’s publication criteria as it currently stands. Therefore, we invite you to submit a revised version of the manuscript that addresses the points raised during the review process.

In agreement with suggestions of Reviewer 2 I feel that the analysis, and possibly the message, of the paper should undergo a really major revision, as present conclusions seem to have limited support from data.

In fact the main message of the paper is that social capital is associated with the risk of cognitive decline. Yet correlations between measures of SC and MMSE are small. Social cohesion has a correlation of .12, while social interaction has a correlation of 0.09. Moreover based on Table 3, it looks like social cohesion is not associated with MMSE in univariate analysis.

The whole hypothesis of the mediation is made uncertain by the cross-sectional design. In fact both low social capital and longer sleep duration might be consequence, rather than predictors, of cognitive decline. In fact the present message of the paper would be that older subjects should sleep less to maintain their social capital, that seems at odds with evidence suggesting an association between shorter sleep time and increased risk of cognitive decline.

As further statistical suggestion, MMSE distribution is probably non-normal. It would be probably better to categorize it and perform a multinomial regression instead of a linear regression analysis. It would be of interest to include in the same model as predictors social capital, sleep time and interaction between the two factors.

We look forward to receiving your revised manuscript.

Kind regards,

Enrico Mossello

Academic Editor

PLOS ONE

Journal Requirements:

3.Thank you for stating the following in the Funding Section of your manuscript:

[The study was supported by the Research and Development Plan of the 13th five-year plan of

Ningxia autonomous region (the major S&T projects.) (grant number 2016BZ02) and the

National Natural Science Foundation of China (grant number 81860599).]

 [The author(s) received no specific funding for this work.]

Reviewers' comments:

Reviewer's Responses to Questions

**Comments to the Author**

1. Is the manuscript technically sound, and do the data support the conclusions?

Reviewer #1: Yes

Reviewer #2: Partly

2. Has the statistical analysis been performed appropriately and rigorously? 

Reviewer #1: Yes

Reviewer #2: I Don't Know

3. Have the authors made all data underlying the findings in their manuscript fully available?

Reviewer #1: Yes

Reviewer #2: Yes

4. Is the manuscript presented in an intelligible fashion and written in standard English?

Reviewer #1: Yes

Reviewer #2: No

5. Review Comments to the Author

Reviewer #1: This study aimed to examine the sleep duration involvement on the positive association between SC and cognitive functioning in aged people. The topic is quite interesting and study design, writing and formatting are well done. Only some points and issues come to my mind as following:

Introduction

Overlay, this section has been prepared orderly. In the last paragraph, authors want to point the relationship between sleep duration and cognitive functions and properly reference to the study which conclude that homocysteine and beta amyloid may mediate the effects of poor sleep on cognition. However, ref 22 (smoking and sleep) here is not related to the main issue and I think it should be removed and instead another relevant ref can be added.

Methods

Line 133:the fifth domain of MMSE is vague. Please refine it.

Sleep duration: I am wondering why the authors did not consider the sleep quality rather than sleep duration. Then, they could easily use the well-known and standard scale, Pittsburgh sleep questionnaire.

Discussion: in the second paragraph which deals with the association between sleep duration and cognition, authors remark the sleep fragmentation and OSA as the possible causations of long sleep duration and low MMSE scales. But, I can’t understand why the authors mention here to the short sleep duration as this is a very definite and old finding and also they have no data on this issue. Rather, more relevant findings regarding the physiological alteration associated with long sleep duration could be of most benefits.

the last paragraph regarding the positive correlation of SC and MMSE scores could be discussed more deeply with mentioning to the some relevant basic findings.

The whole manuscript should be proof read for some grammatical and spelling errors. The example is in Line 210: there no!

Reviewer #2: The article by Wang et al. examines the association between social capital, sleep, and cognitive decline in a sample of older (age > 60 years) adults. Using self-report data, they measure cognitive decline, social capital (via scales measuring social cohesion and social interaction), and sleep duration. Upon conducting a mediation analysis on these variables, they suggest that sleep duration may mediate the connection between social cohesion and cognitive decline. Their study further highlights potential issues both with longer sleep duration and shorter sleep duration.

This is an important topic to investigate, given the increasing evidence supporting a connection between sleep and cognitive impairment in older individuals. Social factors can play a role both in overall physical and mental health. Thus, their connection to the association between sleep and cognitive decline is important to understand. While the overall topic of the article has merit, there are some issues:

Minor issues:

• There are several issues with the overall writing style and appropriate use of grammar. This issue extends throughout the article.

• On line 125, “MMSE” is used for the first time in the text of the article, though the acronym is not explained until the following paragraph. Acronyms should be spelled out upon their first use. This made understanding line 125 a bit difficult.

• Further, the explanation of how the MMSE is scored in the Method section is a bit tricky to decipher. It is unclear if lower scores translate to cognitive impairment, or if higher scores do. This will help with understanding the beginning of the discussion section where the results are summarized.

• On line 194, no r values are given for the association between social cohesion/interaction and MMSE. Only one p-value is provided here.

• On line 221, the author states “SC leads to better cognitive performance…” However, this study cannot provide any causal interpretations of the data since there was no manipulation of variables.

• On line 222, the author states that SC is negatively associated with risk of cognitive decline, but the correlations between social cohesion/interaction and MMSE were positive. Further, on line 251, the author then states that SC is positively correlated with MMSE. These statements should be edited so that they agree with the data.

• In the results, it was unclear why analyses were done both with those who slept less than 8 hours and those who slept 8 hours +. Perhaps a brief discussion of this in the methods or in the results section would help the reader better understand this analysis.

• The paragraph starting on line 226 seems to be making the point that there is evidence that both long and short sleep are connected to cognitive decline. However, the language pointing to this conclusion seems to be a little vague in this paragraph. Specifically, there is a sudden jump from talking about evidence about longer sleep duration to shorter sleep duration. A transition word or sentence might help to bridge this gap and clarify the overall point of this paragraph.

• The paragraph starting on line 242 seems to have a similar issue. The point here appears to be that people with more social capital are busier, and thus are getting less sleep, but this point again seems to be vague and somewhat difficult for the reader to decipher.

Major issues:

• The measurement of sleep duration appears to be problematic. Specifically, I’m not sure how appropriate it is to simply add 1 hour to the overall calculated sleep duration if someone answers “yes” to the question about whether they napped. Ideally, a follow-up question could have asked respondents who indicated napping how long they were asleep during the nap. This kind of information is fairly easy to record, so it is unclear why that was not originally built into the design.

• The measurement of alcohol use seems a bit unclear. Based on the article, it appears that this might have been a yes/no measurement (i.e., they either were or were not an alcohol user). However, the question asks if they’ve had one drink in the last 12 months. If this is the case, it seems as though this question is missing out on quite a bit of variation in use of alcohol. I’m also not sure if it would be appropriate to classify someone who had one drink nearly a year ago as an alcohol user.

• In the results, the author does not mention the fact that correlations between measures of SC and MMSE are small. Social cohesion has a correlation of .12, while social interaction has a correlation of .009. While these may be significant, they are quite small. Recognition of this fact might be appropriate to discuss in the results and discussion sections.

• Based on Table 3, it looks like social cohesion is not associated with MMSE under model 1, but that it is under model 2. However, in the text, the authors say that social cohesion is associated with MMSE for model 1, and that this association remains after controlling for covariates under model 2. Any results given in-text and in tables/diagrams should match.

• Further, because social capital was measured in two ways (i.e., social cohesion and social interaction), results for these two measures might be easier to understand if they are separated in the text. Perhaps focusing on the analysis for cohesion first and then interaction under separate headings would allow the reader to understand the results for both of these measures more easily.

6. PLOS authors have the option to publish the peer review history of their article (what does this mean?). If published, this will include your full peer review and any attached files.

Reviewer #1: **Yes: **Vahid Hajali

Reviewer #2: No

---

## [Author Response · Author response to Decision Letter 0]

13 Jan 2021

Editor comments

1. In agreement with suggestions of Reviewer 2 I feel that the analysis, and possibly the message, of the paper should undergo a really major revision, as present conclusions seem to have limited support from data.

Response: We have modified the manuscript entirely according to reviewer’s suggestion; now we believe it better than the previous version. 

2. In fact the main message of the paper is that social capital is associated with the risk of cognitive decline. Yet correlations between measures of SC and MMSE are small. Social cohesion has a correlation of .12, while social interaction has a correlation of 0.09. Moreover, based on Table 3, it looks like social cohesion is not associated with MMSE in univariate analysis.

Response: Because the research on social capital is biased towards sociology, there are many potential influencing factors, and the effect size is small (usually the effect value of social science is small, but it does not affect the existence of the effect), and this study mainly to explore the mediation effect. In the multivariate analysis, there is a weak correlation between social capital and cognitive decline. The path analysis shows that sleep duration played a mediator role in the relationship between social capital and cognitive decline. 

3. The whole hypothesis of the mediation is made uncertain by the cross-sectional design. In fact both low social capital and longer sleep duration might be consequence, rather than predictors, of cognitive decline. In fact the present message of the paper would be that older subjects should sleep less to maintain their social capital, that seems at odds with evidence suggesting an association between shorter sleep time and increased risk of cognitive decline.

Response: We have discussed our findings more comprehensively; now, it’s easier for the reader to understand the relationship between social capital, sleep duration, and cognitive decline. In this study, we mainly focus on the mediation effect of sleep duration, and the results showed social capital was positively associated with cognitive function and negatively associated with sleep duration; meanwhile, sleep duration was also negatively associated with cognitive function, so long sleep duration might be a risk factor for cognitive function. Furthermore, a longitudinal cohort study [1] reported possessing a rich social capital has been associated with better cognitive function among the old adults. Robbins et al. found lower social capital members in groups were seen for long sleepers [2]. Additionally, as described in a review [3], sleep fragmentation can result in poor quality sleep, depression, and underlying disease processes, such as CHD, which appear to be relevant to the association between long sleep and cognitive function. Reverse causality, the possibility that cognitive function determines sleep duration, cannot be ruled out in our analyses. However, the longitudinal data [4] revealed that sleep duration “increase from 7 or 8 hours” was associated with lower cognitive function scores. Meanwhile, a previous study [5] has also provided no firm evidence that cognitive decline predicts sleep duration. 

[1]. Bennett DA, Schneider JA, Tang Y, Arnold SE, Wilson RS. The effect of social networks on the relation between Alzheimer's disease pathology and level of cognitive function in old people: a longitudinal cohort study. Lancet Neurol. 2006; 5(5): 406-412.

[2] Robbins R , Jean-Louis G , Gallagher R A , et al. Examining Social Capital in Relation to Sleep Duration, Insomnia, and Daytime Sleepiness[J]. Sleep Medicine, 2019.

[3] Grandner MA, Drummond SP. Who are the long sleepers? Towards an understanding of the mortality relationship. Sleep Med Rev 2007;11:341-60.

[4] Ferrie J E , Shipley M J , Akbaraly T N , et al. Change in sleep duration and cognitive function: findings from the Whitehall II Study.[J]. Sleep(5):565-73.

[5] Yaffe K, Blackwell T, Barnes DE, Ancoli-Israel S, Stone KL. Preclinical cognitive decline and subsequent sleep disturbance in older women. Neurology 2007;69:237-42.

4. As further statistical suggestion, MMSE distribution is probably non-normal. It would be probably better to categorize it and perform a multinomial regression instead of a linear regression analysis. It would be of interest to include in the same model as predictors social capital, sleep time and interaction between the two factors.

Response: Yes, as you mentioned, the MMSE score fit a non-normal distribution. Thus, we performed a logarithmic transformation of the MMSE score to finish the mediation effect modeling that requires continuous data, and linear regression employed in the univariate analysis consequently. Besides, categorizing continuous variables may lose much information of the original variable and severely downward precision of the effect estimation.

Reviewer #1

1. Introduction: Overlay, this section has been prepared orderly. In the last paragraph, authors want to point the relationship between sleep duration and cognitive functions and properly reference to the study which conclude that homocysteine and beta amyloid may mediate the effects of poor sleep on cognition. However, ref 22 (smoking and sleep) here is not related to the main issue and I think it should be removed and instead another relevant ref can be added.

Response: Now, we have removed the description of “previous study revealed that sleep duration plays a significant mediating role in the relationship between smoking and mild cognitive impairment [22]” in the last paragraph of introduction and ref 22 (smoking and sleep).

2. Line 133: the fifth domain of MMSE is vague. Please refine it.

Response: We has refined the fifth domain of MMSE as constructional praxis (coping task, eg. copying intersecting pentagons), can be seen in line 132 -133.

3. Sleep duration: I am wondering why the authors did not consider the sleep quality rather than sleep duration. Then, they could easily use the well-known and standard scale, Pittsburgh sleep questionnaire 

Response: The relationship between sleep quality and health has been well studied in the past decades; We believe sleep duration is one dimension of sleep quality, at least among old adults. And more and more research has focused on the relationship between sleep duration and cognitive function; to our knowledge, no study was conducted among Chinses old adults who were living in different cultural backgrounds from other countries. 

4. Discussion: in the second paragraph which deals with the association between sleep duration and cognition, authors remark the sleep fragmentation and OSA as the possible causations of long sleep duration and low MMSE scales. But, I can’t understand why the authors mention here to the short sleep duration as this is a very definite and old finding and also they have no data on this issue. Rather, more relevant findings regarding the physiological alteration associated with long sleep duration could be of most benefits. 

Response: We removed the description of short sleep duration. And there another possible explanation: “Besides, long sleep duration has been associated with an increased level of inflammatory factors and elevated inflammatory cytokine levels increase the risk of cognitive decline” in line 240. 

5. Discussion: the last paragraph regarding the positive correlation of SC and MMSE scores could be discussed more deeply with mentioning to the some relevant basic findings. 

Response: We have extensively edited the discussion section to explain the relationship between SC and cognitive function. See “Furthermore, a recent study manifested that social capital could help older adults continue to independently live in local communities and handle life stressors efficiently, even when they encounter declines in their physical and cognitive health. Additionally, social capital can provide older residents with a sense of security and belonging and is an important reserve domain in old age [54].”

6. The whole manuscript should be proof read for some grammatical and spelling errors. The example is in Line 210: there no! 

Response: We have checked the grammatical and spelling errors entirely by our colleague at Duke University. All the changes are marked in red. 

Reviewer #2 

1. There are several issues with the overall writing style and appropriate use of grammar. This issue extends throughout the article.

Response: We have checked the grammatical and spelling errors entirely by our colleague at Duke University. All the changes are marked in red. 

2. On line 125, “MMSE” is used for the first time in the text of the article, though the acronym is not explained until the following paragraph. Acronyms should be spelled out upon their first use. This made understanding line 125 a bit difficult.

Response: Many thanks for your suggestions. We have spelled out the acronym of MMSE in the last third line of the field process part and marked in red. 

3. Further, the explanation of how the MMSE is scored in the Method section is a bit tricky to decipher. It is unclear if lower scores translate to cognitive impairment, or if higher scores do. This will help with understanding the beginning of the discussion section where the results are summarized.

Response: We have clarified this concern in line 132 in the Methods section; the higher MMSE score reflects the lower cognitive decline levels.

4. On line 194, no r values are given for the association between social cohesion/interaction and MMSE. Only one p-value is provided here.

Response: We have added r values and p-value for the association between social cohesion/interaction and MMSE. See line 195 “both the social cohesion (r=0.12, P<0.001) and social interaction (r=0.09, P<0.05) was positively correlated with MMSE score”.

5. On line 221, the author states “SC leads to better cognitive performance…” However, this study cannot provide any causal interpretations of the data since there was no manipulation of variables.

Response: We have rewritten the sentence as “the positive relationship between SC and better cognitive performance in older Chinese adults was found”, can be seen in line 224-225.

6. On line 222, the author states that SC is negatively associated with risk of cognitive decline, but the correlations between social cohesion/interaction and MMSE were positive. Further, on line 251, the author then states that SC is positively correlated with MMSE. These statements should be edited so that they agree with the data.

Response: As we noted in line 132, a higher MMSE score reflects the lower cognitive decline levels. So SC is positively correlated with MMSE means, which is negatively associated with the risk of cognitive decline.

7. In the results, it was unclear why analyses were done both with those who slept less than 8 hours and those who slept 8 hours +. Perhaps a brief discussion of this in the methods or in the results section would help the reader better understand this analysis.

Response: The average sleep duration of the total participants was 8.1 hours, and we categorized the sleep duration into two groups using a cutoff point of 8 hours. We revised this as “Considering the U-shape relationship between sleep duration and cognitive decline, the exploring analysis conducted stratified (as a cutoff point of mean sleep duration of the total population) by sleep duration showed in Table 5”.

8. The paragraph starting on line 226 seems to be making the point that there is evidence that both long and short sleep are connected to cognitive decline. However, the language pointing to this conclusion seems to be a little vague in this paragraph. Specifically, there is a sudden jump from talking about evidence about longer sleep duration to shorter sleep duration. A transition word or sentence might help to bridge this gap and clarify the overall point of this paragraph.

Response: Same with the editor’s comments 3, now, we have removed the description of short sleep duration.

9. The paragraph starting on line 242 seems to have a similar issue. The point here appears to be that people with more social capital are busier, and thus are getting less sleep, but this point again seems to be vague and somewhat difficult for the reader to decipher.

Response: Due to the U-shape relationship between sleep duration and cognitive function, here our findings support that people with high social activities may keep them in appropriate sleep duration, that means not too short neither too long. Another possible explanation as you mentioned that keep busy may helpful for older adults stay in bed unexpected long time. 

10. The measurement of sleep duration appears to be problematic. Specifically, I’m not sure how appropriate it is to simply add 1 hour to the overall calculated sleep duration if someone answers “yes” to the question about whether they napped. Ideally, a follow-up question could have asked respondents who indicated napping how long they were asleep during the nap. This kind of information is fairly easy to record, so it is unclear why that was not originally built into the design.

Response: We have mentioned in the limitation part as “bedtime and sleep duration were collected via a self-reported survey question; it may involve information bias despite it be commonly used in the epidemiological study due to the feasibility consideration”. In addition, the previous research [1] reported a good correlation when comparing the measurement (subjective and actigraphic measurement) of sleep timing and duration. Furthermore, here the bedtime we asked was the actual sleep onset time, and considering the local life culture, people may like an afternoon nap; we also ask the question ‘Do you sleep at the afternoon?’.

55. Lockley S W, Skene D J, Arendt J. Comparison between subjective and actigraphic measurement of sleep and sleep rhythms. Journal of Sleep Research. 1999; 8:175-183. 

11. The measurement of alcohol use seems a bit unclear. Based on the article, it appears that this might have been a yes/no measurement (i.e., they either were or were not an alcohol user). However, the question asks if they’ve had one drink in the last 12 months. If this is the case, it seems as though this question is missing out on quite a bit of variation in use of alcohol. I’m also not sure if it would be appropriate to classify someone who had one drink nearly a year ago as an alcohol user.

Response: Yes, we have defined the alcohol use as: a drink at least one glass of alcohol (that equal to 1/2 bottle of beer or 125-milliliter grape wine or fruit wine or 40-milliliter white wine) in the past 12 months, seen in line 165-167. In our study, alcohol users include someone who had one drink a year ago, and in the past year.

12. In the results, the author does not mention the fact that correlations between measures of SC and MMSE are small. Social cohesion has a correlation of .12, while social interaction has a correlation of .009. While these may be significant, they are quite small. Recognition of this fact might be appropriate to discuss in the results and discussion sections.

Response: Because the research on social capital is biased towards sociology, there are many potential influencing factors, and the effect size is small (usually the effect value of social science is small, but it does not affect the existence of the effect), and this study mainly explores the mediation effect. In the multivariate analysis, there is a weak correlation between social capital and cognition. The analysis shows that sleep duration is the mediation variable. The interpretability of the analysis results is explained from the perspective of path analysis. We supplemented the description in line 196-197 of results and marked in red.

13. Based on Table 3, it looks like social cohesion is not associated with MMSE under model 1, but that it is under model 2. However, in the text, the authors say that social cohesion is associated with MMSE for model 1, and that this association remains after controlling for covariates under model 2. Any results given in-text and in tables/diagrams should match.

Response: We have revised the description of the tables, and now the results given in text matched the tables. 

14. Further, because social capital was measured in two ways (i.e., social cohesion and social interaction), results for these two measures might be easier to understand if they are separated in the text. Perhaps focusing on the analysis for cohesion first and then interaction under separate headings would allow the reader to understand the results for both of these measures more easily.

Response: We have clarified this as: Social capital is measured with one instrument with two dimensions, social cohesion and social interaction, not two ways, so we analyzed together in our study.

---

## [Decision Letter · Decision Letter 1]

18 Feb 2021

PONE-D-20-28198R1

Social capital and cognitive decline: does sleep duration mediate the association?

PLOS ONE

Dear Dr. Wang,

Thank you for submitting your manuscript to PLOS ONE. After careful consideration, we feel that it has merit but does not fully meet PLOS ONE’s publication criteria as it currently stands. Therefore, we invite you to submit a revised version of the manuscript that addresses the points raised during the review process.

ACADEMIC EDITOR:

I have appreciated the way Authors have refocuesed the discussion. However I would advise to be more cautious whan (in the Abstract) thay state that "improvement of sleep duration may help in maintaining cognitive function", as literally this would be to make people sleep less, which is clearly not the message of the data.

Moreover, I have asked for a statistical revision of the manuscript, as I was not sure of the methodological robustness of findings, due to the small association observed between social cohesion and cognitive function (even non significant in Regression model 1, Table 3), thus making the whole assumptions of a mediation analysis uncertain.

Comments of the reviewer are quite reassuring. Yet I feel that the question regarding how you adjusted the variables in the regression analysis is important (which in the form of quantitative, dichotomous, and categorical), to be sure that the assumptions of the regression model are fulfilled. Moreover, the analyses were adjusted with many variables which could lead to multicollinearity issue.

The reviewer also suggests to use Structural Equation Modelling (SEM) approach to give a clearer picture of the overall pathway direction involving the variables and to validate the findings.

We look forward to receiving your revised manuscript.

Kind regards,

Enrico Mossello

Academic Editor

PLOS ONE

Reviewers' comments:

Reviewer's Responses to Questions

**Comments to the Author**

1. If the authors have adequately addressed your comments raised in a previous round of review and you feel that this manuscript is now acceptable for publication, you may indicate that here to bypass the “Comments to the Author” section, enter your conflict of interest statement in the “Confidential to Editor” section, and submit your "Accept" recommendation.

Reviewer #3: (No Response)

2. Is the manuscript technically sound, and do the data support the conclusions?

Reviewer #3: Partly

3. Has the statistical analysis been performed appropriately and rigorously? 

Reviewer #3: No

4. Have the authors made all data underlying the findings in their manuscript fully available?

Reviewer #3: Yes

5. Is the manuscript presented in an intelligible fashion and written in standard English?

Reviewer #3: Yes

6. Review Comments to the Author

Reviewer #3: The manuscript entitled ‘Social capital and cognitive decline: does sleep duration mediate the association?’ with the aim to examine the mediation effect of sleep duration on the relationship between social capital and cognitive decline in Chinese older adults.

This is quite an interesting study; however, the manuscript requires further improvement.

Comments

Page 8 Line 156-157, adding an hour to the sleep count for someone who took afternoon nap is less accurate without asking the subjects the number of hours.

Page 9 Line 169-170, proper citation for SPSS including publisher name to be stated.

Page 9 Line 165, the definition criteria for alcohol use is inaccurate and could have classified it in a day or in a week for the past 12 months.

Page 9 Line 173, the sentence requires revision.

Page 9 Line 177, for Hayes [28]was, was to be spaced out.

Results

Page 10 Table 1, proper symbol for chi-square to be provided. The symbol chi-square and t and statistical tests which were employed in Table 1 to be stated in the statistical analysis section and denoted in the table footnote.

Page 11 Table 2, an explanation or a note to be provided on how the variables other than continuous/dichotomous variables were adjusted in the partial correlation in the table footnote. The name of correlation coefficient to be stated.

Page 11 Line 197, r=-0.09 to be replaced with r=-0.11

Page 11 Line 200, what ‘multivariate’ refers to in Table 3 to be clearly denoted.

Page 12 Table 3, Model 2 and Model 3 in the table footnote to be written in a new line. Symbol β to be denoted in the table footnote. There were many variables with different type of data/scale of measurement other than continuous/dichotomous variables that were adjusted in the analysis. How these variables were coded and employed in the analysis to be clearly stated (Likewise with Table 4). A note on statistical assumptions fulfillment would be useful. Model summary to be provided.

Page 13 Line 216-219, Table 5, whether the analysis were adjusted or otherwise to be clearly stated.

SEM approach could be explored or as a validation for the regression analysis.

Not all references conformed to the journal format.

7. PLOS authors have the option to publish the peer review history of their article (what does this mean?). If published, this will include your full peer review and any attached files.

Reviewer #3: No

---

## [Author Response · Author response to Decision Letter 1]

8 Mar 2021

ACADEMIC EDITOR:

1. I have appreciated the way the Authors have refocuesed the discussion. However I would advise to be more cautious whan (in the Abstract) thay state that "improvement of sleep duration may help in maintaining cognitive function", as literally this would be to make people sleep less, which is clearly not the message of the data.

Response: We have rephrased the conclusion as " It may be a suggestive clue to identify those at a higher risk of progressing to cognitive impairment.” 

2. Moreover, I have asked for a statistical revision of the manuscript, as I was not sure of the methodological robustness of findings, due to the small association observed between social cohesion and cognitive function (even non significant in Regression model 1, Table 3), thus making the whole assumptions of a mediation analysis uncertain.

Response: Our results found a significant association between social cohesion and cognitive function after adjust the potential covriates (β=0.16, P=0.005) with a effect size (R square) 0.393 (this is a large effect size) , considering the precedently increased number of older adults world widely, and the few chanagable predictors available to reduce the cognitive disorders, our findings provide primary evidence that support comprehensive interventional program in need to maintain the cognitive function in community level and individual level. 

Also, we have performed sensitivity analysis using the SEM, and the results validated the methodological robustness of mediation analysis (Fig 3). And there are statisticians suggested that the significant association of dependent variable with independent variable is unnecessary in the process of mediation analysis (Zhao et al, 2010). 

Zhao X , Jr J G L , Chen Q . Reconsidering Baron and Kenny: Myths and Truths about Mediation Analysis[J]. Journal of Consumer Research, 2010, 37(2):197-206.

3. Comments of the reviewer are quite reassuring. Yet I feel that the question regarding how you adjusted the variables in the regression analysis is important (which in the form of quantitative, dichotomous, and categorical), to be sure that the assumptions of the regression model are fulfilled. Moreover, the analyses were adjusted with many variables which could lead to multicollinearity issue.

Response: In this study, we used the quantitative form of the variable. All descriptions can be seed in line158-169 of pages 8-9. Meanwhile, we conducted the collinearity diagnostics, and the results revealed no multicollinearity among those variables. As shown below, where the tolerance >0.1 and Variance Inflation Factor (VIF) <5 indicate multicollinearity can be ignored (Mason, 1991). 

 The collinearity diagnostics

　 tolerance VIF tolerance VIF

age 0.927 1.078 BMI 0.975 1.026

gender 0.640 1.562 smoking 0.705 1.419

residence 0.702 1.424 alcohol use 0.793 1.262

marital status 0.936 1.069 FPG 0.987 1.013

education 0.619 1.615 hyperlipidemia 0.973 1.028

ethnicity 0.808 1.237 hypertension 0.953 1.050

family income 0.645 1.551 

Dependent variable: MMSE score

4. The reviewer also suggests to use Structural Equation Modelling (SEM) approach to give a clearer picture of the overall pathway direction involving the variables and to validate the findings.

Response: We have performed the SEM analysis. The results as shown in Fig 3. The findings are consistent with the Bootstrap methods of PROCESS.

Fig 3 The path analysis of the mediation effect of sleep duration in the relationship between social cohesion and MMSE 

Reviewers' comments:

1. Page 8 Line 156-157, adding an hour to the sleep count for someone who took afternoon nap is less accurate without asking the subjects the number of hours.

Response: As any self-report measurement, the accuracy of data can not fully be guaranteed. According to our study, most people reported that their afternoon nap time was more or less 1 hour; statistically, it’s reasonable to adjust the total sleep hour by adding one hour to those who report having an afternoon nap.

2. Page 9 Line 169-170, proper citation for SPSS including publisher name to be stated.

Response: We revised the description of SPSS as “Analyses were performed using the Statistical Package for the Social Sciences (SPSS) version 24.0 (IBM Inc., Chicago, Illinois, USA)”.

3. Page 9 Line 165, the definition criteria for alcohol use is inaccurate and could have classified it in a day or in a week for the past 12 months.

Response: We have modified the definition as: a drink of at least one glass of alcohol, that equals 1/2 bottle of beer or 125-milliliter grape wine or fruit wine or 40-milliliter white wine, in a day for the past 12 months. 

4. Page 9 Line 173, the sentence requires revision.

Response: We revised the sentence “A correlation matrix was created using partial correlations under controlling for age, gender, ethnicity, residence, educational attainment, marital status, family income, BMI, smoking, alcohol use, FBG, hyperlipidemia, hypertension” as “Partial correlations were employed to create correlation matrix under controlling for age, gender, ethnicity, residence, educational attainment, marital status, family income, BMI, smoking, alcohol use, FBG, hyperlipidemia, hypertension”.

5. Page 9 Line 177, for Hayes [28]was, was to be spaced out.

Response: we rephrased the sentence as “Bootstrap methods of PROCESS developed by Hayes was employed to test the mediation effect of sleep duration on the relationship between SC and MMSE score [29]”.

6. Page 10 Table 1, proper symbol for chi-square to be provided. The symbol chi-square and t and statistical tests which were employed in Table 1 to be stated in the statistical analysis section and denoted in the table footnote.

Response: We have symbolled the two different tests in table 1 and denoted the symbol of chi-square and t-tests in the Table 1 footnote.

7. Page 11 Table 2, an explanation or a note to be provided on how the variables other than continuous/dichotomous variables were adjusted in the partial correlation in the table footnote. The name of the correlation coefficient to be stated.

Response: We have added a footnote as “The covariates include age, gender, ethnicity, residence, education, marital status, family income, BMI, smoking, alcohol use, FBG, hyperlipidemia, hypertension were adjusted using Partial correlation method.”

8. Page 11 Line 197, r=-0.09 to be replaced with r=-0.11

Response: We have corrected this error. 

9. Page 11 Line 200, what ‘multivariate’ refers to in Table 3 to be clearly denoted.

Response: We have clarified this issue, and denoted in the Table 3 where explained each of the models.

10. Page 12 Table 3, Model 2 and Model 3 in the table footnote to be written in a new line. Symbol β to be denoted in the table footnote. There were many variables with different type of data/scale of measurement other than continuous/dichotomous variables that were adjusted in the analysis. How these variables were coded and employed in the analysis to be clearly stated (Likewise with Table 4). A note on statistical assumptions fulfillment would be useful. Model summary to be provided. 

Response: in Table 3, Model 2 and Model 3 in the table footnote have already been written in a new line and Symbol β was denoted in the table footnote. Additionally, variables other than continuous/dichotomous variables that were marital status and family income, we coded them in page 16 of measurement section. Model summary was provided in Statistical Analyses section as well as denoted in the table 3.

Three separate linear regression models were performed to examine the association among SC, sleep duration, and MMSE. In summary, model l include the SC and sleep duration; model 2 adjusted with covariate variables (age, gender, ethnicity, residence, education, marital status, family income, body mass index, smoking, alcohol use, fasting blood glucose, hyperlipidemia, hypertension); and the interaction between SC and sleep duration were added in model 3. Social interaction and social cohesion were tested separately in all the models.

11. Page 13 Line 216-219, Table 5, whether the analysis were adjusted or otherwise to be clearly stated.

Response: Yes, the analysis were adjusted covariate variables, and we supplemented the description “After controlling for age, gender, ethnicity, residence, education, marital status, family income, BMI, smoking, alcohol use, FBG, hyperlipidemia, hypertension.” In the table 5 footnote.

12. SEM approach could be explored or as a validation for the regression analysis.

Response: We peformed the SEM approach to validate the findings, can be seen in sensitivity analysis section and the results showed in Fig 3. 

13. Not all references conformed to the journal format.

Response: We have formatted the reference list carefully; now it’s more suitable for the journal style.

---

## [Decision Letter · Decision Letter 2]

29 Apr 2021

PONE-D-20-28198R2

Social capital and cognitive decline: does sleep duration mediate the association?

PLOS ONE

Dear Dr. wang,

Thank you for submitting your manuscript to PLOS ONE. After careful consideration, we feel that it has merit but does not fully meet PLOS ONE’s publication criteria as it currently stands. Therefore, we invite you to submit a revised version of the manuscript that addresses the points raised during the review process.

We look forward to receiving your revised manuscript.

Kind regards,

Y Zhan

Academic Editor

PLOS ONE

Journal Requirements:

Reviewers' comments:

Reviewer's Responses to Questions

**Comments to the Author**

1. If the authors have adequately addressed your comments raised in a previous round of review and you feel that this manuscript is now acceptable for publication, you may indicate that here to bypass the “Comments to the Author” section, enter your conflict of interest statement in the “Confidential to Editor” section, and submit your "Accept" recommendation.

Reviewer #3: (No Response)

2. Is the manuscript technically sound, and do the data support the conclusions?

Reviewer #3: Partly

3. Has the statistical analysis been performed appropriately and rigorously? 

Reviewer #3: (No Response)

4. Have the authors made all data underlying the findings in their manuscript fully available?

Reviewer #3: Yes

5. Is the manuscript presented in an intelligible fashion and written in standard English?

Reviewer #3: Yes

6. Review Comments to the Author

Reviewer #3: The authors have put in great effort to address the comments.

Minor clarification/revision required.

Line 160, for marital status (unmarried, married and widowed/divorced), was the variable collapsed (0.1) for the regression analysis?

Line 161- 161, it was mentioned that family income <1,000 RMB,162 1,000-1,999 RMB, 2,000-2,999 RMB, 3,000-4,999 RMB, and 5,000 RMB or more were collected. For Table 1, was the family income variable coded as <1000 and ≥ 1000 (0 and 1)? If so, which one was used in the regression analysis; (0.1) or scale data?

Line 200, likewise for the partial correlation analysis, marital status and family income to be clearly denoted.

Education to be revised to Educational attainment (years).

7. PLOS authors have the option to publish the peer review history of their article (what does this mean?). If published, this will include your full peer review and any attached files.

Reviewer #3: No

---

## [Author Response · Author response to Decision Letter 2]

1 May 2021

Dear editor:

We are submitting the revised manuscript titled “Social capital and cognitive decline: does sleep duration mediate the association?” for publication. We have modified the manuscript along the lines suggested by editors and reviewers. In addition, we re-checked the reference and now it is complete and correct. All the changes are marked in red.

Below are our responses to the reviewer’s comments point by point.

1. Line 160, for marital status (unmarried, married and widowed/divorced), was the variable collapsed (0.1) for the regression analysis?

Response: We devided marital status into two categories (married vs. unmarried/widowed/divorced) in all the statistical analysis due to the few reponders are unmarried (10 of the 955). And we have running the analysis procedure again, all the changes marked in red. 

2. Line 161- 161, it was mentioned that family income <1,000 RMB,162 1,000-1,999 RMB, 2,000-2,999 RMB, 3,000-4,999 RMB, and 5,000 RMB or more were collected. For Table 1, was the family income variable coded as <1000 and ≥ 1000 (0 and 1)? If so, which one was used in the regression analysis; (0.1) or scale data?

Response: The value of family income was ordinal variable in the regression model (divided into five degree: <1,000 RMB,162 1,000-1,999 RMB, 2,000-2,999 RMB, 3,000-4,999 RMB, and 5,000 RMB or more). We have also rehearched it in statistical analysis part as “family income (ordinal)”, marked in red.

3. Line 200, likewise for the partial correlation analysis, marital status and family income to be clearly denoted.

Response: As described above, We have revised the marital status and family income as marital status (married vs. not married (included unmarried and widowed/divorced)), family income (ordinal)”, now it is more clearly.

---

## [Editor Report · Decision Letter 3]

12 May 2021

Social capital and cognitive decline: does sleep duration mediate the association?

PONE-D-20-28198R3

Dear Dr. wang,

We’re pleased to inform you that your manuscript has been judged scientifically suitable for publication and will be formally accepted for publication once it meets all outstanding technical requirements.

Kind regards,

Y Zhan

Academic Editor

PLOS ONE
---

## [Editor Report · Acceptance letter]

18 May 2021

PONE-D-20-28198R3 

Social capital and cognitive decline: does sleep duration mediate the association? 

Dear Dr. Wang:

I'm pleased to inform you that your manuscript has been deemed suitable for publication in PLOS ONE. Congratulations! Your manuscript is now with our production department. 

Kind regards, 

on behalf of

Dr. Y Zhan 

Academic Editor

PLOS ONE